

# Cloning and expression analysis of cinnamoyl-CoA reductase (CCR) genes in sorghum

Jieqin Li[*], Feifei Fan[*], Lihua Wang, Qiuwen Zhan, Peijin Wu, Junli Du, Xiaocui Yang and Yanlong Liu

College of Agriculture, Anhui Science and Technology University, Fengyang, China
[*] These authors contributed equally to this work.

## ABSTRACT

Cinnamoyl-CoA reductase (CCR) is the first enzyme in the monolignol-specific branch of the lignin biosynthetic pathway. In this research, three sorghum CCR genes including *SbCCR1*, *SbCCR2-1* and *SbCCR2-2* were cloned and characterized. Analyses of the structure and phylogeny of the three CCR genes showed evolutionary conservation of the functional domains and divergence of function. Transient expression assays in *Nicotiana benthamiana* leaves demonstrated that the three CCR proteins were localized in the cytoplasm. The expression analysis showed that the three CCR genes were induced by drought. But in 48 h, the expression levels of *SbCCR1* and *SbCCR2-2* did not differ between CK and the drought treatment; while the expression level of *SbCCR2-1* in the drought treatment was higher than in CK. The expression of the *SbCCR1* and *SbCCR2-1* genes was not induced by sorghum aphid [*Melanaphis sacchari* (Zehntner)] attack, but *SbCCR2-2* was significantly induced by sorghum aphid attack. It is suggested that *SbCCR2-2* is involved in the process of pest defense. Absolute quantitative real-time PCR revealed that the three CCR genes were mainly expressed in lignin deposition organs. The gene copy number of *SbCCR1* was significantly higher than those of *SbCCR2-1* and *SbCCR2-2* in the tested tissues, especially in stem. The results provide new insight into the functions of the three CCR genes in sorghum.

## INTRODUCTION

Lignin is a complex aromatic polymer present mainly in the secondary cell walls of vascular plants. It plays an important role in specialized conducting and supporting tissues of plants, facilitating water transport, providing mechanical strength, and defending against biotic and abiotic stress (*Bhuiyan et al., 2009*; *Vermerris, Sherman & McIntyre, 2010*; *Jin et al., 2014*). The lignin biosynthetic pathway has attracted researchers' attention because lignin is a limiting factor in a number of agro-industrial processes, such as chemical pulping, forage digestion, and the conversion of lignocellulosic plant biomass to bioethanol (*Poovaiah et al., 2014*; *Tang et al., 2014*).

Lignin is derived from the polymerisation of monomeric subunits known as monolignols. The three main monolignols in grasses are *p*-coumaryl, coniferyl and sinapyl alcohols, which

Corresponding author
Jieqin Li, lijq@ahstu.edu.cn, wlhljq@163.com

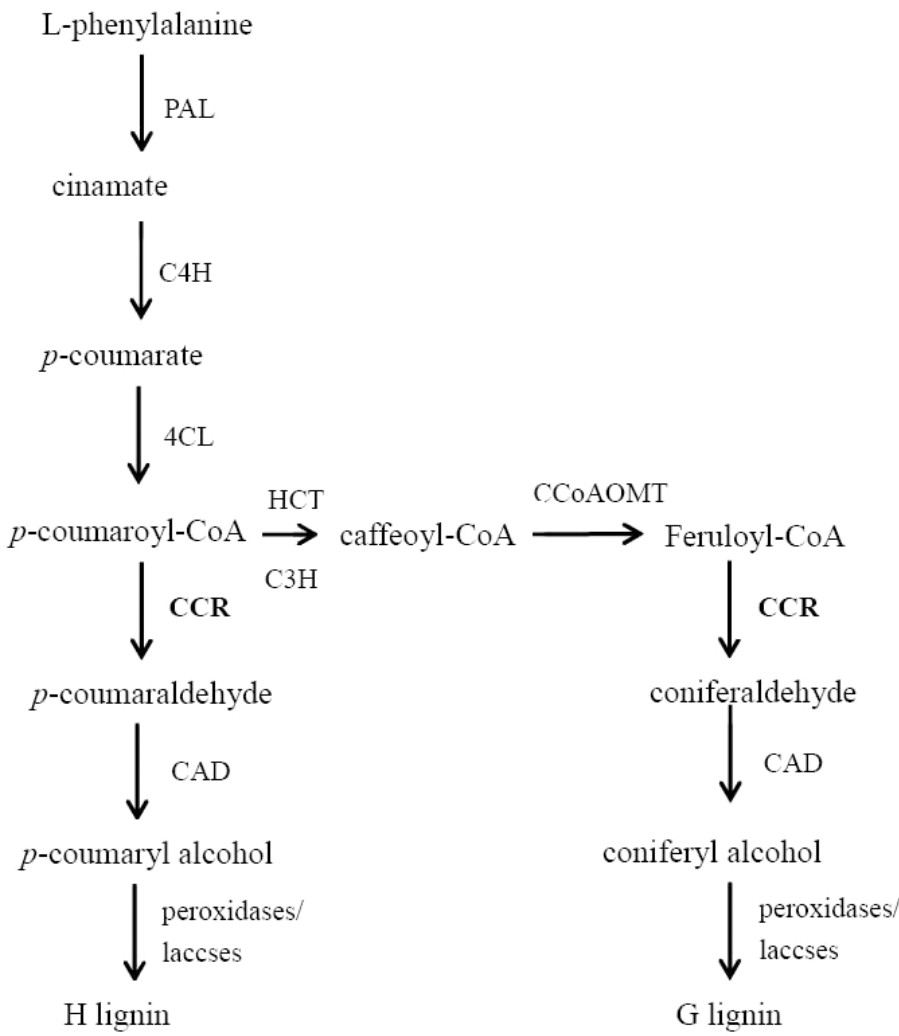

**Figure 1** **The outline of the lignin biosynthetic pathway.** Abbreviations: *PAL*, phenylalaine ammonia-lyase; *C4H*, cinnamate 4-hydroxylase; *4CL*, 4-coumarate: CoA ligase; *HCT*, *p*-hydroxycinnamoyl-CoA: quinate shikimate *p*-hydroxycinnamoyltransferase; *C3H*, *p*-coumarate 3-hydroxylase; *CCoAOMT*, caffeoyl-CoA *O*-methyltransferase; *CCR*, cinnamoyl-CoA reductase; *CAD*, cinnamyl alcohol dehydrogenase.

give rise to *p*-hydroxyphenyl (H), guaiacyl (G) and syringyl (S) residues, respectively (*Boerjan, Ralph & Baucher, 2003*; *Vermerris, Sherman & McIntyre, 2010*). The biosynthesis of monolignols requires the action of a series of enzymes.

Cinnamoyl-CoA reductase (CCR) is the first enzyme in the monolignol-specific branch of the lignin biosynthetic pathway, where it converts feruloyl-CoA to coniferaldehyde (Fig. 1) (*Leple et al., 2007*). Genes encoding CCR proteins have been studied in many species including Arabidopsis (*Lauvergeat et al., 2001*; *Xue et al., 2015*), rice (*Kawasaki et al., 2006*), tobacco (*Chabannes et al., 2001*), soybean (*Luderitz & Grisebach, 1981*), poplar tree (*Leple et al., 2007*), maize (*Pichon et al., 1998*; *Tamasloukht et al., 2011*), dallisgrass (*Giordano et al., 2014*) and wheat (*Ma & Tian, 2005*). The down regulation of CCR in transgenic Arabidopsis

**Table 1** The primers used in the research.

| Primer name | Forward primer | Reverse primer | Usage |
|---|---|---|---|
| GSbCCR1 | CATCACTCGACCGCACATAC | CAGCCAGCGAACAAACACTA | |
| GSbCCR2-1 | TCAGACTAATAACCCGCCTAG | GCAGTATCAGCGTTGGAAA | Gene cloning |
| GSbCCR2-2 | TCTTTCCGCTTCCACCGAT | GACTGCCAAAATTTAATAACCAA | |
| SSbCCR1 | CGGACTAGTATGACCGTCGTCGAC | TCCCCCGGGCGCACGGATGGCGAT | Subcellular localization |
| SSbCCR2-1 | CGGACTAGTATGCCAACAGCAGAG | CGCGGATCCTGATTTGTGGAGTTG | |
| SSbCCR2-2 | CGGACTAGTATGGCCGTCGTCGTG | TCCCCCGGGAAGTTTTGAAATCAA | |
| YSbCCR1 | ATGCTGCTCGAGAAGGGATACAC | GTTCTTCGGGTCATCTGGGTTC | |
| YSbCCR2-1 | ACCCACAAAGTGCAAGGACGAC | CCGCTGGTTCGTGAACTTGTATCC | qRT-PCR |
| YSbCCR2-2 | GGAGTACCCTATTCCGACAAGGTG | GCACTGGCGTGAACTTGATTCC | |

and tobacco leads to significant reduction of lignin content. The CCR gene family is very diverse in plants (*Barakat et al., 2011*). Multiple homologs of CCR genes involved in different functions can be present in a same plant. For instance, *AtCCR1* is involved in developmental lignification, while *AtCCR2* is for stress and elicitor response (*Lauvergeat et al., 2001*). It has been demonstrated that different CCR genes play different roles in plant's development. Although many studies have been done on CCR genes in different plants, to date, little work on CCR genes in sorghum has been done.

In this study, three CCR genes were cloned from the cDNA in sorghum seedlings by homology cloning. The structures and functions of their deduced proteins were analyzed by a bioinformatics method. To understand the expression of the three CCR genes in different tissues and treatments, expression analyses were performed. The findings provide a basis for further revealing the roles of the three CCR genes in sorghum.

## MATERIALS AND METHODS

### Plant materials and gene cloning

Sorghum Tx623B was used for gene cloning and expression analysis. Seeds of Tx623B were pretreated with 75% alcohol for 2 min and then washed three times with distilled water. The sterilized seeds were planted in pots. After germination, the seedlings were grown at 28 °C under 12 h light/12 h dark for two weeks, taken and ground in liquid $N_2$ for RNA extraction. Total RNA of the seedling samples was extracted using an RNA Prep Pure Plant kit (Tiangen Co., Beijing, China) and was reverse transcribed using a SuperScript II kit (TaKaRa Biomedical, Shiga, Japan).

Cloning primers for the three CCR genes (i.e., *Sb07g021680* named *SbCCR1*, *Sb02g014910* named *SbCCR2-1*, *Sb04g005510* named *SbCCR2-2*) were designed according to the CCR gene sequences in the National Center for Biotechnology Information (NCBI) database. The amplification primers are listed in Table 1. Polymerase chain reaction (PCR) was performed using KOD FX polymerase (Toyobo, Osaka, Japan). The amplified fragments were separated on 1% agarose gels and purified by DNA Gel Extraction Kit (AxyPrep DNA Gel Extraction kit; Axygen, USA). Then, the purified fragments were linked to pMD18-T vector (Takara, China) and transformed into *E. coli* DH5α. The recombinant

plasmids were verified by PCR; the positive clones were sent to Genscript Company (Nanjing, China) for sequencing.

## Drought and pest defense experiments

For the drought defense experiment, Tx623B seeds were sterilized as described above and germinated on wet paper. After germination, the seedlings were grown at 28 °C under 12 h light/12 h dark for 2 weeks. Then half of the seedlings were treated with 20% polyethylene glycol (PEG) 6000 as drought treatment and sampled at 24 h and 48 h. The other half of the seedlings were untreated with PEG 6000 but sampled at the same times as control.

For the pest defense experiment, Tx623B seeds were sterilized, germinated and grown as in the drought defense experiment. Then, for half of the seedlings, 4 pest-attacked sorghum leaves with 2–4 sorghum aphids [*Melanaphis sacchari* (Zehntner)] on each were put close to each seedling so that the aphids would migrate to the seedlings; subsequently, the seedlings were placed in a cage. After the sorghum seedlings were attacked by the aphids, their pest-attacked leaves were sampled at 24 h and 48 h for RNA extraction. The other half of the seedlings was used as control and leaves were sampled at the same times for RNA extraction.

## Sequence analysis and phylogenetic analysis

Nucleotide sequences were translated into protein sequences which were then aligned using BioEdit. Phylogenetic analyses were performed using the neighbor joining (NJ) method with Mega 5.1 software. Branch support was assessed with 1,000 bootstrap replicates.

## Subcellular localization

The primer pairs for the three CCR genes were used to amplify the cDNA fragments encoding the full-length CCR proteins. The PCR fragments for each CCR gene were inserted into the vector 1305GFP at the N-terminus of the green fluorescent protein (GFP) under the control of cauliflower mosaic virus 35S promoter. An *Agrobacterium tumefaciens* strain carrying the *35S::CCR1-GFP*, *35S::CCR2-1-GFP*, *35S::CCR2-2-GFP* or *35S::GFP* plasmid was infiltrated into *Nicotiana benthaminana* leaves and analyzed with confocal microscopy 48 h after agroinfiltration as described previously (*Goodin et al., 2002*). Fluorescence of GFP was observed with a Leica LSM710 confocal laser scanning microscope.

## Quantitative real-time PCR

Quantitative real-time PCR (qRT-PCR) was performed using a SYBR Green supermix (Bio-Rad, USA) on an ABI prism 7900 real-time PCR system. The primers for the three CCR genes are listed in Table 1. The sorghum *eIF4a1* gene (*Sb04g003390*) was used as the endogenous control in the experiment. All reactions were run in three replicates. The $2^{-\triangle\triangle CT}$ method was used to analyze relative changes in gene expression (*Livak & Schmittgen, 2001*).

## Absolute qRT-PCR

The three genes' coding fragments were aligned with pMD18-T vector. Fresh 10-fold serial dilutions were created from the three constructs. The concentration of the plasmid preps was measured using a Biodrop Touch spectrophotometer. The corresponding gene copy

**Table 2  Molecular characteristics of CCR genes in *Sorghum bicolor*.**

| Name | Gene number | CDS length | Peptide residue | Theoretical Mw (kDa) | Theoretical PI |
|------|-------------|-----------|-----------------|----------------------|----------------|
| SbCCR1 | Sb07g021680 | 1,549 | 374 | 40.24 | 5.47 |
| SbCCR2-1 | Sb02g014910 | 1,268 | 346 | 38.21 | 6.50 |
| SbCCR2-2 | Sb04g005510 | 1,378 | 343 | 37.95 | 8.79 |

number in plasmid was calculated with the following equation:

$$\text{gene copy number} = \frac{6.02 \times 10^{23} (\text{copy/mol}) \times \text{DNA amount (g)}}{\text{DNA length (bp)} \times 660 \left(\frac{g}{\text{mol}}/\text{bp}\right)}$$

where DNA length stands for the combined length of plasmid and insert, in base pairs, and DNA amount equals the corresponding plasmid concentration times the volume (*Gotia et al., 2016*). The linear regression equation and $R^2$ were obtained by comparing the log values of the DNA concentration and the Ct values. Then, gene copy number was calculated with the corresponding linear regression equation.

## RESULTS

### Cloning and structure analysis of *SbCCR1*, *SbCCR2-1* and *SbCCR2-2*

Based on data obtained from the sorghum genome sequence and research results (*Giordano et al., 2014*), 3 CCR genes were identified by PCR and sequencing. Sequence annotation and protein chemical characterization of the three CCR genes are shown in Table 2. The sequencing results showed that the cDNA sequences of the three CCR genes were completely consistent with those in NCBI database. Molecular analysis of the deduced proteins showed that SbCCR1, SbCCR2-1 and SbCCR2-2 contained 374, 346 and 343 amino acids, respectively. Multiple alignments showed that the highest similarity between SbCCR1 and ZmCCR1 was 89.3% (Fig. 2). The similarity between SbCCR2-1 and SbCCR2-2 (59.8%) was higher than that between SbCCR1 and SbCCR2-1 (58.3%) and that between SbCCR1 and SbCCR2-2 (49.3%).

Commonly, a CCR protein contains an NADP binding domain and a catalytic activity domain (NWYCY) (*Pan et al., 2014*). The multiple alignments showed that SbCCR1, SbCCR2-1 and SbCCR2-2 contained an NADP binding domain, an NWYCY domain and a substrate domain (Fig. 2). These results demonstrated that the three CCR genes contained the functional domains of CCR, suggesting that the three CCR proteins should be of CCR activities.

### Phylogenetic analysis of the CCR proteins

Phylogenetic analysis of the CCR proteins encoded in sorghum and other plant genomes showed that there were two major clades (Fig. 3). One clade was mainly comprised of CCR1 proteins; the other was mainly comprised of CCR2 proteins. The two clades were supported with moderate bootstrap values. SbCCR1 was included in the first clade; while SbCCR2-1 and SbCCR2-2 were included in the second clade. The clade containing mainly CCR1 proteins was comprised of two major sub-clades: one consisted of dicotyledons (e.g.,

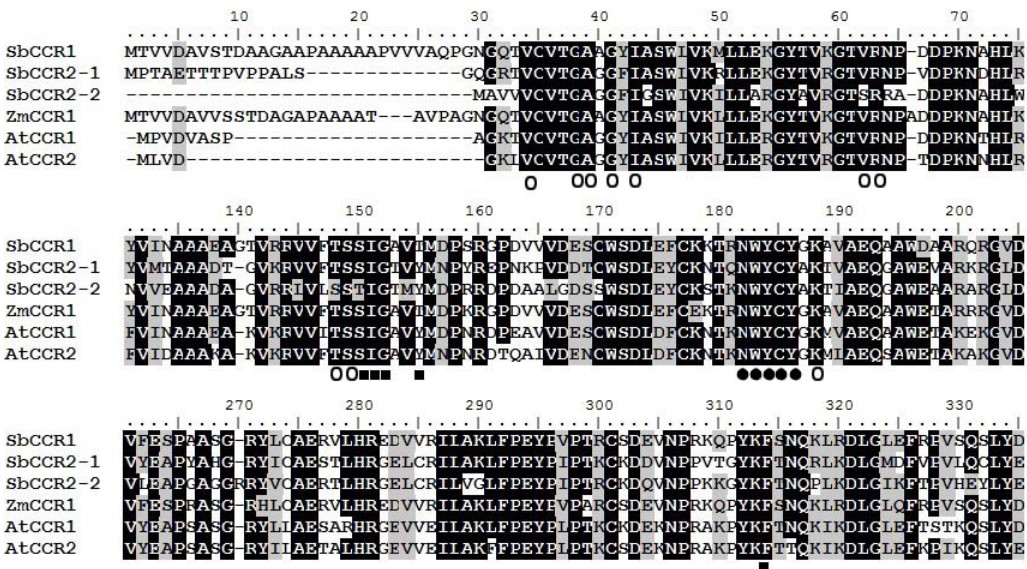

**Figure 2** **Multiple alignment of the protein sequences of sorghum CCR proteins and other plant CCR proteins.** The alignment includes sequences from *Arabidopsis* (*AtCCR1* and *AtCCR2*) and maize (*ZmCCR1*). The amino acids believed to be part of the active sites are shown below the alignment with the following codes: ●, catalytic activity domain; ○, NADP binding domain; ■, substrate binding domain.

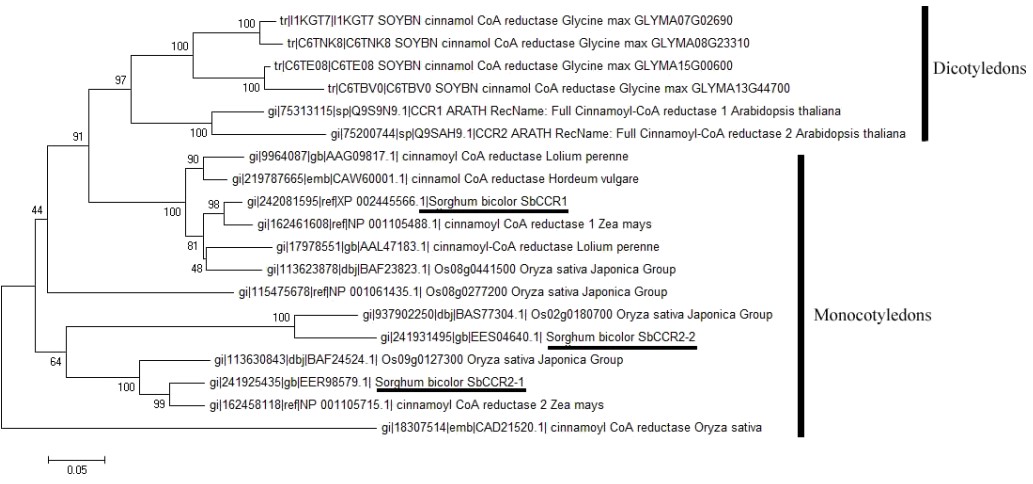

**Figure 3** **Phylogenetic analyses of sorghum CCR proteins and other plant CCR homologues.**

*Glycine max* and *Arabidopsis thaliana*), the other monocotyledons (e.g., *Sorghum bicolor*, *Hordeum vulgare*, *Lolium perenne*, *Zea mays* and *Oryza sativa*) and both were supported with high bootstrap values.

## Subcellular localization of CCR proteins

CCR genes perform their functions in cytoplasm. Therefore, the three CCR proteins were further analyzed with the protein subcellular localization prediction program WoLF PSORT (http://wolfpsort.org/). All of these proteins were predicted to be in cytoplasm. To verify

the predicted results, a transient expression assay for the three CCR proteins was performed in *N. benthamiana* leaves. GFP alone was expressed as control, SbCCR1, SbCCR2-1 and SbCCR2-2 in full-length were fused to the N-terminus of GFP. The results showed that free GFP dispersed throughout the cytoplasm in the *N. benthamiana* epidermal cells; and the green fluorescent signal of GFP was not co-located with the auto-fluorescence of chlorophylls in chloroplasts. The localization patterns of SbCCR1-GFP, SbCCR2-1-GFP and SbCCR2-2-GFP were similar to that of free GFP (Fig. 4), indicating that the three CCR proteins were targeted to the cytoplasm.

## Expression analysis of CCR genes in drought and pest treatments

To understand the functions of the three CCR genes in sorghum, expression analysis was performed in drought treatment (Fig. 5). The results showed that the expression levels of *SbCCR*1, *SbCCR2-1* and *SbCCR2-2* were significantly higher in the drought treatment than in CK at 24 h, indicating that all of the three CCR genes were induced by drought. But at 48 h, there were no differences in the expression levels of *SbCCR1* and *SbCCR2-2* between CK and the drought treatment; the expression level of *SbCCR2-1* was higher in the drought treatment than in CK. The results suggested that all of these genes were involved in the process of drought defense. *SbCCR1* and *SbCCR2-2* mainly responded in the first phase of drought defense, while the *SbCCR2-1* gene responded during the whole drought defense period.

To identify the functions of the three CCR genes in pest defense, expression analysis was performed. The expression levels of the *SbCCR1* and *SbCCR2-1* genes were not different between CK and the pest treatment at 24 h or 48 h (Fig. 6). But the expression levels of the *SbCCR2-2* gene at 24 h and 48 h in the pest treatment were significantly higher than in CK, indicating that *SbCCR2-2* was involved in pest defense.

## Spatial expression patterns of CCR genes

The linear regression equations and $R^2$ values of the three CCR genes were obtained by comparing the log values of DNA concentration and the Ct values. The slopes of the standard quantification curves for *SbCCR1, SbCCR2-1* and *SbCCR2-2* were −3.25, −3.56 and −3.39, respectively. The $R^2$ values of the three equations were 0.99. The gene copy numbers of *SbCCR1*, *SbCCR2-1* and *SbCCR2-2* in various plant organs, including leaf, spikelet, stem and root, were quantified using their corresponding linear regression equations. The results demonstrated that the three CCR genes were expressed in allthe tested organs (Fig. 7). The expression patterns of the three CCR genes were consistent with the deposition patterns of lignin in these organs. The highest expression levels of the three CCR genes were found in stem. The expression levels of *SbCCR1* and *SbCCR2-2* in stem and root were obviously higher than in spikelet and leaf, but those of *SbCCR2*-1 in stem and spikelet were higher than in leaf and root. The gene copy number of *SbCCR1* was significantly higher than those of *SbCCR2-1* and *SbCCR2-2* in root, leaf, stem and spikelet, especially in stem. This indicated that the expression level of *SbCCR1* was obviously higher than those of *SbCCR2-1* and *SbCCR2-2* in the tested sorghum tissues.

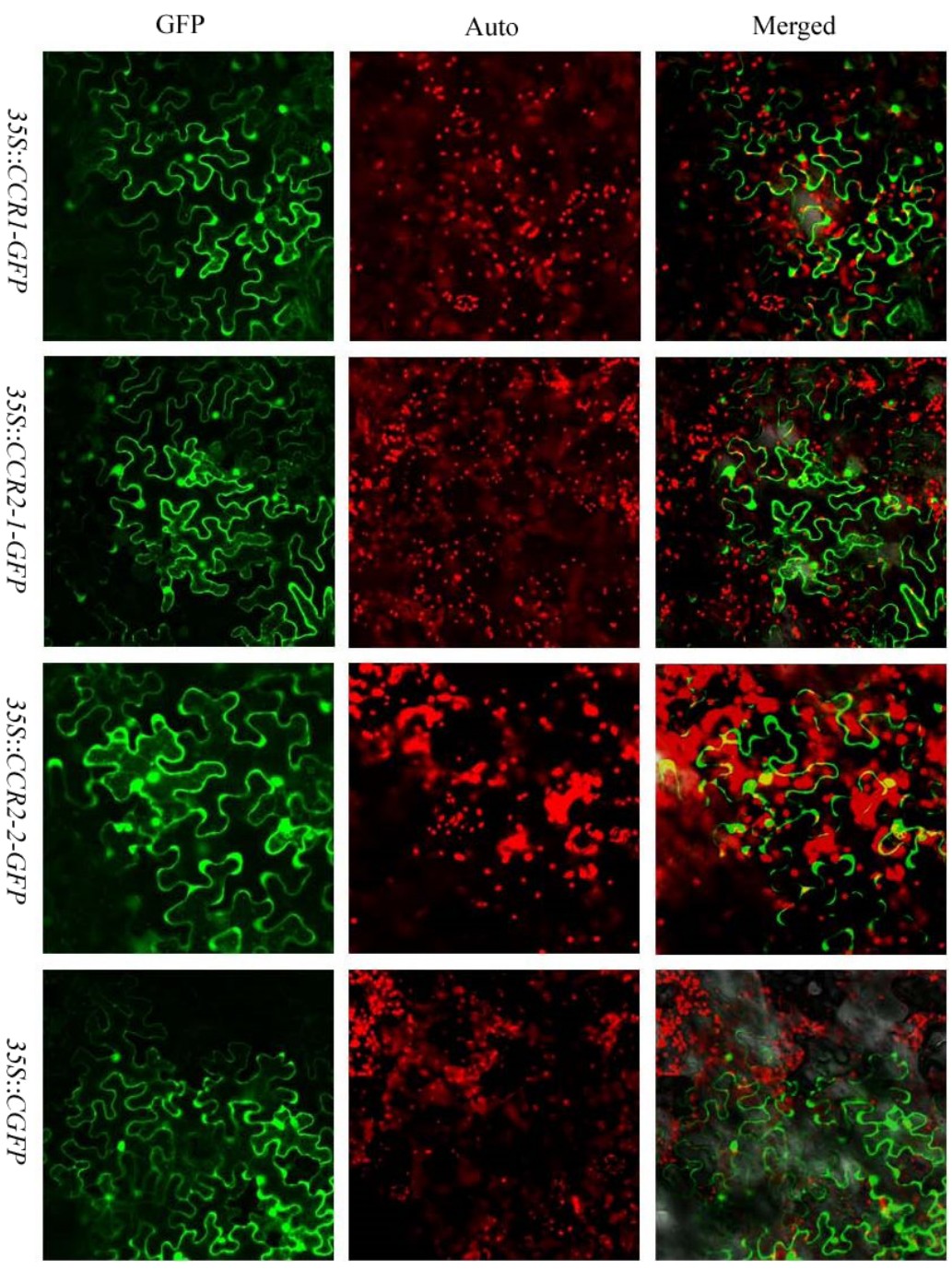

**Figure 4  The subcellular localization of sorghum CCR genes in *N. benthamiana* leaves.** Transient expression of SbCCR1–GFP, SbCCR2-1-GFP and SbCCR2-2-GFP fusion proteins and GFP in *N. benthamiana* leaves; GFP, fluorescence of SbCCR1-GFP, SbCCR2-1-GFP, SbCCR2-2-GFP and GFP; Auto, Chl auto-fluorescence; Merged, merged images of GFP and Auto ones in bright.

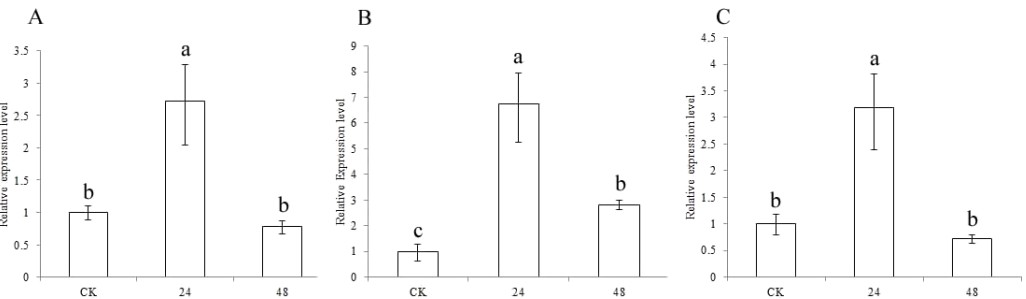

**Figure 5    Expression of sorghum CCR genes in CK and the drought treatment at different times.** (A), *SbCCR1*; (B), *SbCCR2-1*; (C), *SbCCR2-2*; variance analysis was used for statistical test; different lowercase letters represent significant difference at 0.05 probability level.

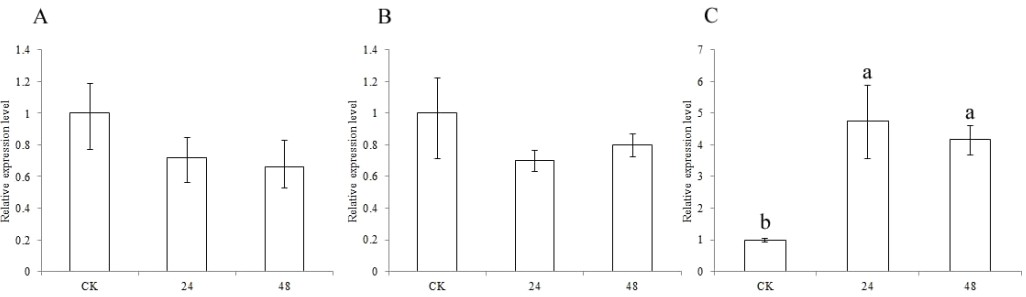

**Figure 6    Expression of sorghum CCR genes in CK and the pest treatment at different times.** Pest used was sorghum aphids (*Melanaphis sacchari* (Zehntner)); A, *SbCCR1*; B, *SbCCR2-1*; C, *SbCCR2-2*; variance analysis was used for statistical test; different lowercase letters represent significant difference at 0.05 probability level.

## DISCUSSION

The biosynthesis of lignin begins with the common phenylpropanoid pathway starting with the deamination of phenylalanine and leading to cinnamoyl-CoA esters which are then channeled into the lignin branch pathway via CCR (*Pichon et al., 1998*). CCR genes were first cloned and characterized in *Eucalyptus*, providing insight into the regulation of the lignin biosynthetic pathway in developmental and defense processes (*Lacombe et al., 1997*). In this research, the molecular and expression characteristics of three CCR genes in sorghum were reported. The results showed that the three CCR genes exhibited different biochemical properties and differential expression patterns in response to drought and pest treatments.

In higher plants, there are a number of CCR and CCR-like genes (*Barakat et al., 2011*). But only one to two CCR genes (bona fide CCR genes) are involved in lignin biosynthesis during plant development. For example, *AtCCR1* is the only bona fide CCR gene in *Arabidopsis* (*Mir et al., 2008*). In this study, the *SbCCR1* gene showed high sequence similarity with the *ZmCCR1* gene which is the bona fide CCR gene in maize (*Tamasloukht et al., 2011*). Phylogenetic analysis revealed that SbCCR1 was closer to other CCR1 proteins involved in lignin biosynthesis in plant developmental processes. The expression analysis also showed that the *SbCCR1* gene was mainly expressed in stem where lignin was deposited

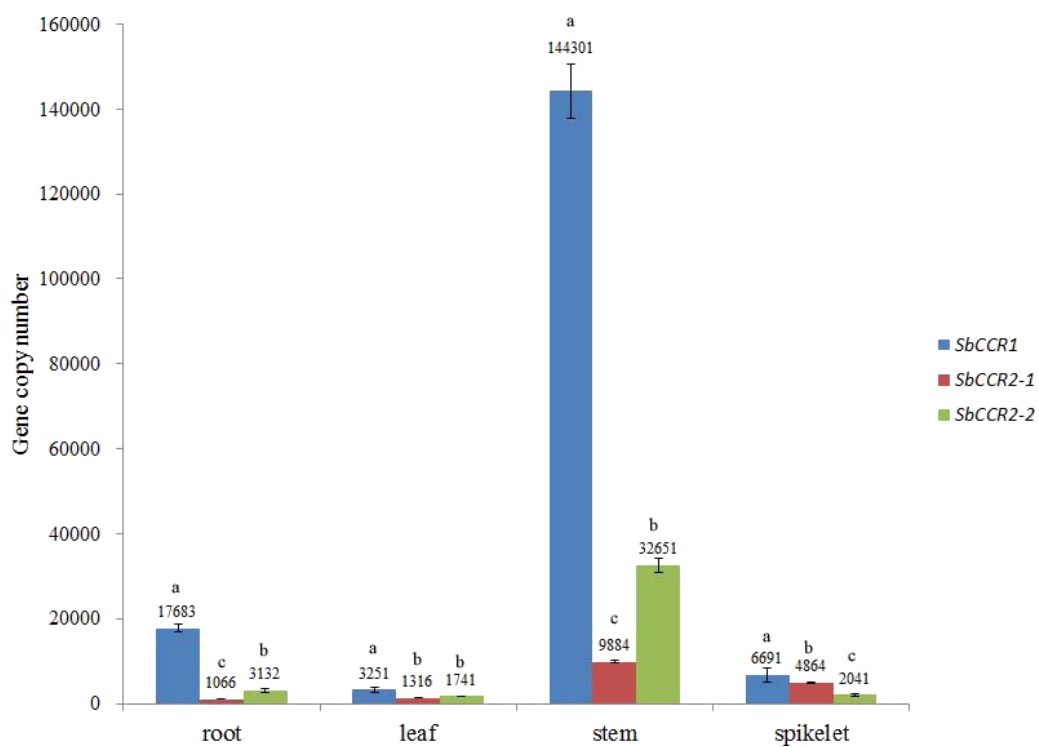

**Figure 7 The copy numbers of CCR genes in different sorghum tissues.** Variance analysis was used for statistical test; different lowercase letters represent significant difference at 0.05 probability level.

at high content. Additionally, the expression level of *SbCCR1* was significantly higher than those of *SbCCR2-1* and *SbCCR2-2* in various tissues. This indicated that the *SbCCR1* gene should be mainly involved in the biosynthesis of lignin. All these results indicated that *SbCCR1* should be a bona fide CCR gene in sorghum.

The CCR2 gene is mainly involved in defense-related processes in plant (*Fan et al., 2006*). In this research, phylogenetic analysis revealed that *SbCCR2-1* and *SbCCR2-2* were in a same clade, indicating that the two genes should belong to CCR2 gene. Spatial expression analysis showed that the three CCR genes were mainly expressed in tissues with lignin deposited, but expression analysis of the CCR genes in drought and pest treatments showed that *SbCCR2-1* was mainly involved in the process of drought defense, while *SbCCR2-2* was involved in pest defense. All these results suggested that *SbCCR2-1* and *SbCCR2-2* mainly played a role in defense-related processes in sorghum.

In plants, the CCR1 and CCR2 genes show different expression patterns in response to biotic and abiotic stresses (*Fan et al., 2006*; *Tamasloukht et al., 2011*). For instance, *AtCCR1* and *AtCCR2* show different expression patterns in response to pathogen infection. *AtCCR1* transcript level is not significantly changed, but AtCCR2 highly accumulates in infected leaves with *Xanthomonas campestris* pv. *campestris* (*Lauvergeat et al., 2001*). In this research, it was also observed that the transcript levels of *SbCCR1* and *SbCCR2-1* were not significantly affected by pest. Whereas, the transcript level of *SbCCR2-2* was strongly affected by pest

attack. It was showed that *SbCCR2-1* and *SbCCR2-2* played different roles in response to pest attack, suggesting a role of *SbCCR2-2* in pest resistance.

## CONCLUSIONS

In this research, three CCR genes were cloned and characterized in sorghum. Phylogenetic analysis revealed that the *SbCCR1* gene belongs to one clade, while *SbCCR2-1* and *SbCCR2-2* belong to another clade. Subcellular localization indicated that the three CCR genes were localized in the cytoplasm. qRT-PCR demonstrated that the three CCR genes were induced by drought. But the expression pattern of *SbCCR2-1* was different from those of *SbCCR1* and *SbCCR2-2*. The expressions of the *SbCCR1* and *SbCCR2-1* genes were not induced by pest attack, but that of *SbCCR2-2* was significantly induced by pest. It was suggested that *SbCCR2-2* was involved in the process of pest defense. Absolute qRT-PCR analysis showed that the three CCR genes have similar spatial expression patterns. The gene copy number of *SbCCR1* was significantly higher than those of *SbCCR2-1* and *SbCCR2-2* in the tested tissues, especially in stem. The results from this study provide important information for understanding the roles of the three CCR genes in sorghum.

### Funding

This work was supported by grants from the young talent project of Anhui province (gxyqZD2016217), the key project of natural science research of Anhui provincial education department (KJ2016A177), the National Natural Science Foundation of China (31301383), the Key-construction Subject Plan of Anhui Province (No: WanJiaoMiKe[2014]28), University outstanding youth talent support program (No: WanJiaoMiRen[2015]211), and the youth talent support program of Anhui Science and Technology University (No:XiaoRenFa[2015]69). The funders had no role in study design, data collection and analysis, decision to publish, or preparation of the manuscript.

### Grant Disclosures

The following grant information was disclosed by the authors:
Anhui province: gxyqZD2016217, WanJiaoMiKe[2014]28.
Anhui provincial education department: KJ2016A177.
National Natural Science Foundation of China: 31301383.
University outstanding youth talent support program: WanJiaoMiRen[2015]211.
Anhui Science and Technology University: XiaoRenFa[2015]69.

### Competing Interests

The authors declare there are no competing interests.

### Author Contributions

- Jieqin Li conceived and designed the experiments, analyzed the data, contributed reagents/materials/analysis tools, wrote the paper.
- Feifei Fan performed the experiments, contributed reagents/materials/analysis tools.

- Lihua Wang and Junli Du performed the experiments.
- Qiuwen Zhan conceived and designed the experiments, reviewed drafts of the paper.
- Peijin Wu and Xiaocui Yang performed the experiments, prepared figures and/or tables.
- Yanlong YL.L Liu analyzed the data.

## Data Availability

The raw data was supplied as a Supplemental Information.

## Supplemental Information

Supplemental information for this article can be found online at http://dx.doi.org/10.7717/peerj.2005#supplemental-information.

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
