# Peer review of "Cloning and expression analysis of cinnamoyl-CoA reductase (CCR) genes in sorghum"

_PeerJ, doi:10.7717/peerj.2005_

## Round 0.1 · original submission · Minor Revisions

I think you should address point #1 (under Validity of the findings) of reviewer 1, i.e. re qPCR. If you can make some attempt to address points 2 and 4 of reviewer 1, that would be good - but if not, please explain why not in your cover letter, and temper your conclusions - i.e. expression analyses are just that, you cannot infer function. You should address all the other comments of both reviewers.

Reviewer 1 ·

Basic reporting

This article reports on the characterization of three CCRs cloned from Sorghum. The authors analyze gene expression patterns of these CCRs in different tissues and during biotic and abiotic stresses. They describe the phylogenetic relationships of the Sorghum CCRs and explore subcellular localization.

1/ Unfortunately, the authors omit citing some relevant literature. Especially when discussing the protein sequence and domain conservation of the Sorghum CCRs, the authors should relate to the findings of Pan et al. (2014), which go into the details of structure-function relationships in CCRs. Also, it was found that CCRs can also contribute to the production of phenylpropene volatile organic compounds (Muhlemann et al., (2014)), not just to lignin formation.
2/ While the text is written clearly, there are a few places where the English could be improved. For example, line 47: “walls of secondary thickened cells” probably should read as “secondary cell walls”.
3/ Line 188 and other places in the text: The authors did not perform a spatio-temporal analysis, only a spatial analysis (different tissues were analyzed). The text should be changed accordingly.
4/ In the discussion, the authors often use the term “real CCR gene”. All CCR genes are real. I suggest that the authors use a more appropriate and scientific term for the group of CCR genes they consider “real”.

Experimental design

1/ The research question(s) and relevance of the project are not clearly stated in the introduction. Why is it important to characterize CCRs in sorghum? What is the knowledge gap that the authors wanted to fill? Characterizing an already well-characterized gene function in another species (Sorghum in this case) is merely an incremental gain of knowledge.
2/ What is the reasoning behind the drought and aphid-stress experiments? Is Sorghum particularly prone to these stresses. If not, why did the authors focus on these two stresses only?
3/ It is difficult to understand the methods section on the induction of biotic stresses with aphids. This section could be improved. Also, did the authors make sure that the aphids migrated from the aphid-infected leaves to the seedlings?

Validity of the findings

1/ I would like to see absolute levels of expression for the qPCR results section. It is possible that one of the CCRs is expressed at much higher levels than the others and thus presumably is the one CCR that contributes the most to formation of monolignols in the given tissue/stress.
2/ Line 77: “To understand the functions of the three CCR genes, expressions…”. Since expression analysis only provides a hint at possible functions, I would have liked to see Sorghum transgenic lines with downregulated CCRs. These transgenic lines would provide a better understanding of CCR function than only expression analysis. Sorghum can be readily transformed by particle bombardment (see e.g. Liu and Godwin, (2012)).
3/ Did the authors make sure that the aphids migrated from the aphid-infected leaves to the seedlings?

4/ CCRs act on multiple hydroxycinnamoyl-CoA esters. I would have liked seeing the substrate specificities and kinetic parameters of the three CCRs, as this could answer whether the different CCRs differentially direct the fluxes to the different monolignols.

·

Basic reporting

Line 59. Please introduce a new Figure 1 with the lignin biosynthetic pathway, highlighting the position of the CCR genes.

Experimental design

Methods are well described and according to the standards in the field.

Validity of the findings

Captions in Figures 5 and 6 must indicate the statistical test used for means comparison.

Additional comments

Line 145-146. The similarity between SbCCR2-1 and SbCCR2-2 was higher (?%) than that between SbCCR1 and SbCCR2-1 (?%) and that between SbCCR1 and SbCCR2-2 (?%).

Fig.2 must highlight in some way the sorgum CCRs and the different CCR1 sub-clades (monocot/dicot).

Fig. 6 and corresponding text (at least in line 215) must indicate the common and scientific name of the pest species used in the study.

---

## Round 0.2 · Minor Revisions

In the summary you state "The gene copy number of SbCCR1 was significantly higher than those of SbCCR2-1 and SbCCR2-2 in the tested tissues, especially in stem" - must be incorrect - the gene copy number does not vary in different tissues, expression levels does. You have it correctly stated on p. 11, lines 10-11. Please use the same phrasing in the summary.

---

## Round 0.3 · accepted · Accept

Your paper is now accepted.